# Eco-Friendly, Low-Cost, and Flexible Cotton Fabric for Capacitive Touchscreen Devices Based on Graphite

Fahad Alhashmi Alamer * and Wedad Aqiely

Department of Physics, Faculty of Applied Science, Umm AL-Qura University, Al Taif Road,
Makkah 24382, Saudi Arabia
* Correspondence: fahashmi@uqu.edu.sa

**Abstract:** Cotton fabrics with high electrical conductivity were prepared using graphite dispersed in ethanol as the conductive material. The graphite particles were drop-cast onto the cotton fabrics at room temperature. The samples were characterized by SEM, EDX, XPS, and XRD. In addition, the electrical properties of the cotton samples were investigated using a four-probe technique. The concentration of the dispersed graphite was increased to a saturation concentration of 74.48 wt% to investigate the relation between the sheet resistance of the conductive cotton and the graphite concentration. With increasing graphite concentration, the sheet resistance decreased and reached the minimum value of 7.97 $\Omega/\square$ at a saturation concentration of 74.48 wt%. Samples with low, medium, and high graphite concentration showed semiconducting metallic behavior at a transition temperature of 90 °C. Based on their individual electrical properties, a smart glove was fabricated for touchscreen devices such as cell phones and self-service devices by dropping a small amount of dispersed graphite into one of the fingertips of the glove. The smart glove showed high efficiency and durability up to 10 wash cycles.

**Keywords:** cotton fabrics; dispersed graphite; semiconducting metallic; smart gloves; touchscreen





## 1. Introduction

The topic of electronic textiles has recently become a multidisciplinary research area due to recent advances in scientific research and technological innovation [1–3]. An ideal candidate for the fabrication of smart textiles is cotton fabrics endowed with electrical conductivity [4–10]. The most important features of these smart textiles are their ability to sense, recognize, and respond to the wearer's movements and behaviors [11–14]. The ability to compute, activate, and transform data are other features of smart textiles [15–18]. Potential applications of smart textiles include wearable sensors [19–21], wireless communication [22], and photovoltaic devices [23]. Various materials such as metals [24], conductive polymers [8–10,25,26], and carbon-based materials [5–7] can be used to fabricate conductive textiles, which is usually performed by simple methods such as casting, spraying, coating, and dipping.

Recently, carbon-based materials such as graphite [27], graphene [28,29], graphene oxide [30,31], reduced graphene oxide [32], and carbon nanotubes [33–35] have attracted great interest in the field of electronic textiles due to their excellent electrical properties and mechanical stability. In the study presented by Alamer et al., highly electrically conductive fabrics were fabricated using single-walled carbon nanotubes (SWCNTs) [36]. The filtration method was used to fabricate the conductive cotton fabric. The results showed that the conductivity of the treated cotton fabrics depended on the concentration of SWCNTs, and a low sheet resistance of 0.006 $\Omega/\square$ was achieved at a concentration of 41.5 wt%. The electrical conductivity of the treated cotton fabrics was affected by temperature, with the conductivity decreasing with increasing temperature, and a transition occurred at about 75 °C, resulting in an increase in conductivity with increasing temperature.

Multiwalled carbon nanotubes (MWCNTs) were also used to fabricate conductive cotton fabrics [37], where a sheet resistance of 15.92 $\Omega/\square$ was achieved, and the fabric exhibited semiconducting behavior over a wide temperature range. Although the use of carbon nanotubes to fabricate conductive fabrics yields high electrical conductivity, there are some drawbacks in their use; for example, SWCNTs are expensive, are difficult to purify, and disperse in gas or liquid. Moreover, the presence of impurities, insolubility, concerns about toxicity, and the tendency of CNTs to cluster are some of the disadvantages of their use. Graphene and their derivative have also been used as a conductive material to make conductive fabrics due to its excellent electrical and mechanical properties [36]. In the study presented by Zhou et al. [28], graphene-based conductive fabrics were fabricated to monitor human motion. The dip-coating technique was used for the fabrication process. They found that the electrical resistance of the conductive fabric decreased with increasing concentration of graphene nanoparticles. They found that the conductive fabrics could be used as strain sensors to detect human movements. Karim et al. [37] fabricated conductive and flexible fabrics from reduced graphene oxide using the pad-dry method. The results showed that the treated fabrics exhibited about 60% improvement in tensile strength. Furthermore, these conductive fabrics can be used as strain sensors for monitoring human activities. In a recent study presented by Xiong et al. [38], graphene textiles were also fabricated and used for human motion monitoring. The results showed that the graphene textile is capable of powering electronic devices due to its high maximum power of 3.6 μW. Recently, the effect of the synthesis method of graphene oxide on the electrical properties of conductive cotton fabrics was investigated [39]. It was found that the conductive cotton fabric prepared by electrochemically synthesized graphene oxide exhibited low sheet resistance compared with the graphene oxide synthesized by Hummers' method.

In this study, the use of graphite to develop electronic textiles based on cotton fabric substrates is investigated. Graphite is composed of carbon layers with covalent and metallic bonds within each layer, with the atoms in each layer arranged in a hexagonal pattern and the layers stacked in order AB [40]. Graphite has good electrical and thermal properties within the layers but poor electrical and thermal properties perpendicular to the layers, which is why graphite is called anisotropic [41]. Graphite is an abundant, nontoxic, and environmentally friendly material that is also one of the most efficient and cost-effective electrical conductors. Another important property of graphite is that it is stable and does not oxidize when washed [42]. It is worth noting that there are a few studies in the literature that have investigated the effects of incorporating graphite into fabrics' physical properties compared with other conductive materials.

In our previous work, cotton fabrics interspersed with graphite were prepared based on the polar solvents dimethyl sulfoxide, dimethyl formamide, and a mixture of both [43]. A brush coating drying process was used to produce the conductive fabrics. The results showed that the best performance was obtained with a minimum sheet resistance of 1.197 $k\Omega/\square$ and a lower graphite content of 58.70 wt% for the cotton fabrics prepared with the graphite dispersion from the mixture of both solvents. Moreover, these conductive cotton fabrics were used to fabricate strain sensors with excellent reproducibility. Woltornist et al. treated conductive PET fabrics by infusing them with a few graphene/graphite layers using an interfacial trapping method and without using modified additives [27]. The electrical conductivity of the treated fabrics was improved, with sheet resistance at 7.4 wt% graphite reaching the minimum value of 3.6 $k\Omega/\square$. In another study [44], a certain amount of graphite was mixed with a polyaniline solution, and the mixture was then applied to the textile using a model doctor blade method. The results showed that the resistance of the conductive textile depended on the thickness of the coating and had minimum values of about 4 to 10 $k\Omega/10$ cm for the sample with thick coating. It was also observed that the resistance increased with an increasing number of washing cycles. Schal et al. [45] have prepared conductive fabrics based on a mixture of graphite and polyurethane. First, the graphite–polyurethane dispersion was prepared using graphite with different particle sizes, and then the mixture was applied to four types of fabrics, namely cotton, polyester, viscose,

and linen, using a doctor's knife method. The resistance was investigated depending on the number of washing cycles (up to 10 washing cycles). It was found that the resistance of polyester and viscose fabrics significantly increased with an increasing number of washing cycles, while the increase was much smaller for cotton and almost negligible for linen. In an interesting study, Alonso et al. [46] fabricated conductive polyester fabrics based on graphite using the back-knife coating technique. The main objective of this study was to investigate the flammability. They found that the polyester fabric with low graphite content did not pass the flammability test, while the polyester fabric with high graphite content passed this test.

In the present study, we report a simple method for producing graphite-based cotton fabrics by incorporating graphite nanopowder dispersed in ethanol into the fabric. The main features of our conductive cotton fabric are as follows: (i) It has a minimum sheet resistance of 7.97 $\Omega/\square$ at a graphite concentration of 74.48 wt%. To our knowledge, this is the lowest sheet resistance reported for a conductive fabric made of graphite nanopowder. (ii) The resistance as a function of temperature shows a semiconductor–metal transition for low, medium, and high graphite concentrations. Based on the distinct electrical properties, we produce smart gloves for touchscreen devices, in which a small amount of graphite dispersion has been used as a conductive material in the fingertips of the gloves to establish electrical contact between our smart gloves and touchscreen devices.

## 2. Materials and Methods

### 2.1. Materials

In this particular study, graphite powder and ethanol were obtained from Sigma-Aldrich and used without any purification process. A pure woven cotton fabric was purchased from a local store.

### 2.2. Preparation of Graphite Dispersion

To prepare graphite dispersion in ethanol, about 100 mg of graphite was added to 4 mL of ethanol and then magnetically stirred for 15 min at room temperature. This procedure was repeated four times using different amounts of graphite, namely 200 mg, 250 mg, 300 mg, and 350 mg.

### 2.3. Synthesis of Conductive Cotton Fabric

All cotton fabrics used for the preparation of the conductive samples had the same area of 1 in$^2$. Figure 1 shows a schematic diagram of the preparation of a graphite-based conductive cotton fabric, including the preparation of the graphite solution. The conductive cotton fabrics were prepared by drop-casting graphite solution into the untreated cotton substrate. Prior to this, the untreated cotton fabric was immersed in deionized water for 5 min to improve the uptake of the graphite dispersion solution. For the first drop-casting cycle, 1 mL of the graphite solution was dropped into the cotton sample and allowed to soak for 10 min to achieve an average moisture removal of 100%. The fabric was then dried in a drying oven at 100 °C for 30 min. Subsequent cycles were performed up to fourteen times using the same method until the saturation concentration was reached to increase the amount of graphite in the sample.

The manufacturing process was inexpensive due to the cheapness of our raw materials, namely graphite, cotton, and ethanol. To calculate the price of our product for a conductive cotton fabric, we need to add the cost of graphite, cotton fabric, and ethanol. According to Sigma-Aldrich, the price of 1 kg of graphite is USD 77, and this amount is enough to make a large number of samples because the amount of graphite in each sample does not exceed 100 mg (this is for samples with a high concentration). Moreover, to make the smart gloves, we add only a small amount of graphite to one of the fingertips of the glove, which means that this process will be inexpensive.

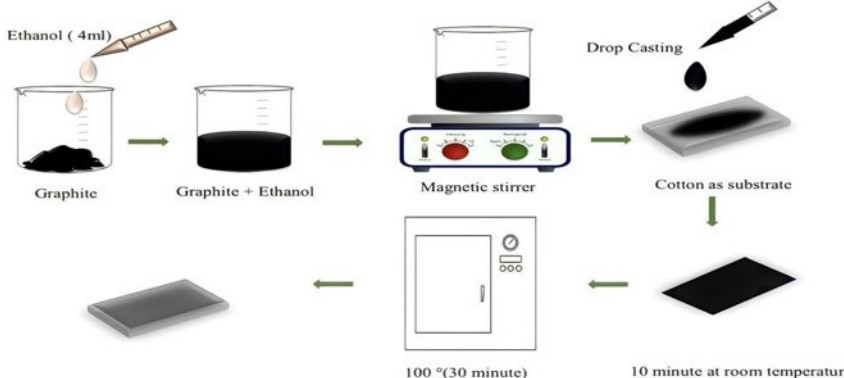

**Figure 1.** Schematic diagram of the preparation of a graphite-based conductive cotton fabric, including the preparation of the graphite solution.

*2.4. Characterizations*

The surface morphology and elemental composition of the untreated and treated cotton fabrics were examined using a Thermo Scientific Scios 2 SEM at 350× and 750× magnification. The XRD spectra of the samples were analyzed with an X-ray diffractometer (A Bruker D8 ADVANCE XRD) using Cu-Kα radiation. X-ray photoelectron spectroscopy (XPS) was performed with a VG-Microtech Multilab electron spectrometer using Mg-Kα (1253.6 eV) radiation. Electrical measurements of the samples were made using a four-probe technique [47] and were performed at a temperature and relative humidity of 24 °C and 65%, respectively. The current from the outer leads was applied with a Keithley 2400 source meter, and the potential difference from the inner leads was measured with an HP 34,401 A multimeter. The I–V curve was used to calculate the electrical resistance of the sample. The sheet resistance in units of ohms per square of the conductive sample was then calculated using the relation $R_S = R (w/L)$, where R is the measured resistance, w is the sample width (2.5 cm), and L is the distance between probes (0.35 cm). The washing test was performed with a steam washing machine (Babyliss, China; V = 220 V−240 V; frequency ~50–60 Hz; power = 1260–1500 W) and without using detergent. The steam washing procedure was repeated for 10 cycles, with each cycle lasting 20 min.

## 3. Results and Discussion

*3.1. Electrical Studies*

To investigate the electrical properties of graphite-based conductive cotton, the sheet resistance of the sample was studied as a function of the amount of graphite as shown in Figure 2A and Table 1. As the amount of graphite increased, the sheet resistance of the conductive sample decreased. The sheet resistance of the sample was 95.96 kΩ/□ at a low concentration of 15.20 wt%. However, when the concentration was increased from 18.00 to 35.02 wt%, the sheet resistance decreased from 35.76 kΩ/□ to 10.05 kΩ/□. A significant decrease in sheet resistance to 83.74 Ω/□ was observed when the graphite concentration increased to 52.66 wt%, and a slight change in sheet resistance values occurred thereafter. The minimum sheet resistance was reached at 7.97 Ω/□ when the graphite concentration was 74.48 wt%. This low value could be due to the fact that the more graphite was added to the fabric, the more interconnections could be made between the graphite and cotton matrix as the number of conducting paths increased. To our knowledge, this is the lowest sheet resistance reported for graphite-based conductive cotton. There was a significant change in the color of the sample from white color for untreated cotton to black color for the highly concentrated sample.

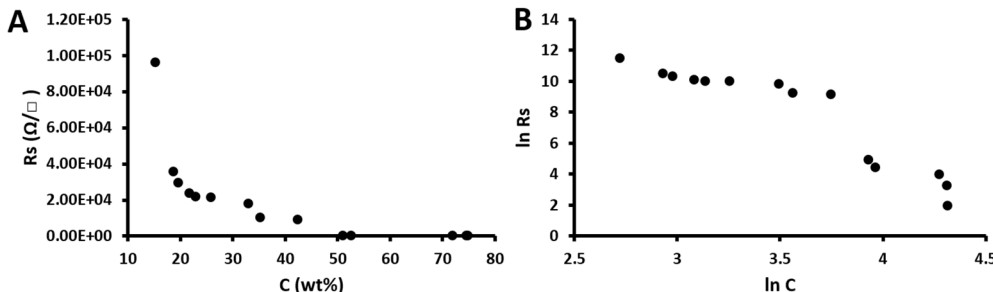

**Figure 2.** (**A**) Sheet resistance of graphite-based cotton at different concentration of graphite. (**B**) Plot between natural logarithm of sheet resistance and natural logarithm of concentration.

**Table 1.** Sheet resistance values of the conductive cotton fabrics at different concentration of graphite.

| C (wt%) | Rs (Ω/□) |
| --- | --- |
| 15.20 | $95.96 \times 10^3$ |
| 18.73 | $35.76 \times 10^3$ |
| 19.64 | $29.37 \times 10^3$ |
| 21.80 | $23.76 \times 10^3$ |
| 23.00 | $21.84 \times 10^3$ |
| 25.92 | $21.36 \times 10^3$ |
| 33.00 | $18.05 \times 10^3$ |
| 35.20 | $10.05 \times 10^3$ |
| 42.39 | $9.15 \times 10^3$ |
| 51.02 | 136.36 |
| 52.66 | 83.74 |
| 71.96 | 53.89 |
| 74.54 | 25.99 |
| 74.84 | 7.97 |

Figure 2A shows a hyperbolic curve, and at a high graphite concentration, it is not clear what the value of the sheet resistance is. To determine the relationship between the sheet resistance and the graphite concentration, we plotted the natural logarithm of the sheet resistance and natural logarithm of the graphite concentration (see Figure 2B). It is obvious that the sheet resistance is inversely proportional to the graphite concentration $\left(R_s \propto \frac{1}{C}\right)$.

To confirm the electrical behavior of the graphite-based cotton fabrics, we investigated the resistance as a function of the sample temperature at low, medium, and high concentrations in the temperature range from room temperature to 130 °C (see Figure 3). Resistance was first measured from the I–V curve at room temperature, then the sample was heated to 40 °C and resistance was measured. Afterward, the sample was heated by 10 °C up to 130 °C, and the resistance was measured each time. As shown in Figure 3A–C, the samples showed the same electrical behavior as a function of temperature. The resistance of the sample decreased with increasing temperature in the range from room temperature to below 90 °C, indicating a semiconductor behavior. At the transition at 90 °C, the resistance increased, indicating a metallic behavior. In our previous study, it was reported that the resistance as a function of temperature showed semiconductor behavior only when the conductive cotton was prepared with graphite dispersed in DMF, DMSO, and a mixture of both [43]. This proves that graphite dispersed in ethanol affects the electrical behavior of the conductive cotton. This semiconductor–metal behavior is a well-known

phenomenon that has been observed in microelectronics, particularly in the fabrication of resistors with carbon-filler paste on a ceramic substrate. In the study presented by Zha et al. [48] on the dependence of resistance on temperature, the semiconductor–metal behavior was observed for the composite with hybrid filler carbon nanotubes–carbon black and polyethylene–polyvinylidene fluoride. They found that the temperature transition depends on the concentration of the filler and the melting point of the polymer. In addition, this behavior is also observed in the manufacture of conductive fabric using conductive polymers, graphene, and carbon nanotubes. Alamer et al. [28,49] demonstrated that the conductive cotton fabrics coated with PEDOT:PSS exhibited semiconductor–metal behavior, and the transition temperature depended on the PEDOT:PSS concentration. In another study [27], the thermal investigation of the graphene/graphite infused PET fabrics showed that the conductive fabrics exhibited semiconductor–metal behavior at 350 K in which the study was conducted in a temperature range from 10 K to 400 K.

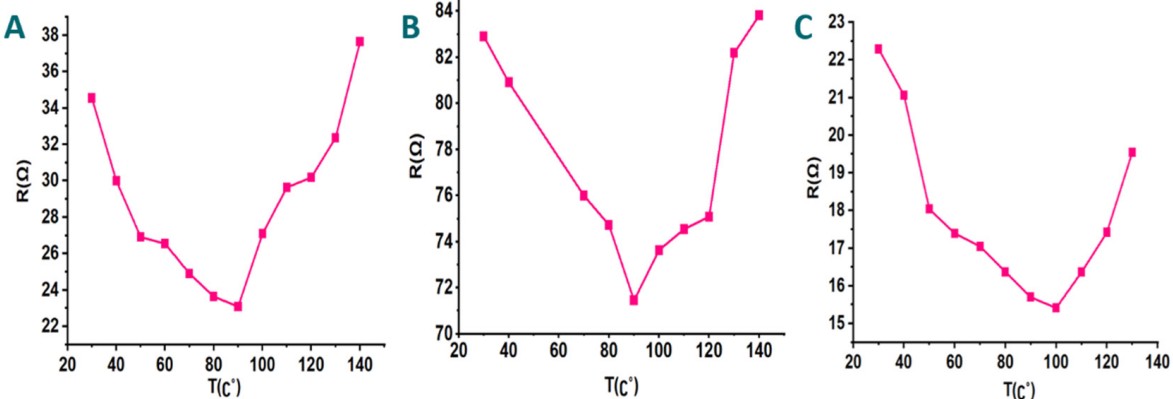

**Figure 3.** Resistance of graphite-based cotton as a function of temperature; (**A**) low concentration of 20 wt.%; (**B**) medium concentration of 30 wt.%; and (**C**) high concentration of 78.78 wt.%.

### 3.2. Morphological Study

The morphology of the untreated and graphite-based cotton was studied using SEM, as shown in Figure 4. The untreated cotton had a smooth and clean surface with long fibers that had an irregular or flat cross-section. In addition, porous spaces between fibers were also observed. However, with increasing graphite concentration, the graphite particles adhered to the fibers and were also located in the pore spaces between the fibers.

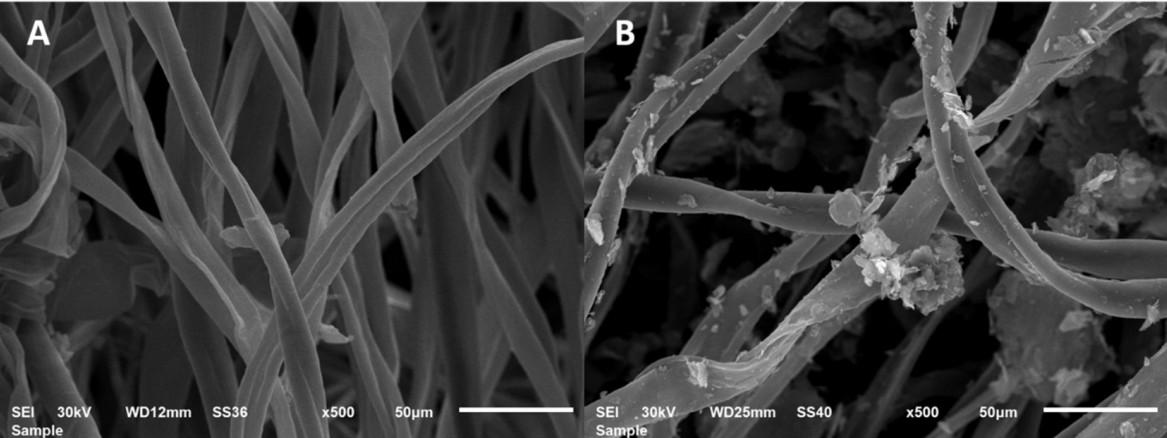

**Figure 4.** Morphology of untreated cotton (**A**) and graphite-based cotton (**B**).

### 3.3. Elemental Studies

### 3.3.1. EDX Analysis

EDX analysis of the untreated and graphite-based cotton was performed to determine the elemental composition of the materials (see Figure 5). The EDX spectra of both samples show peaks attributed to carbon (CK$\alpha$ 0.27 keV) and oxygen (OK$\alpha$ 0.52 keV), which can be attributed to the cellulose structure and graphite in the treated sample.

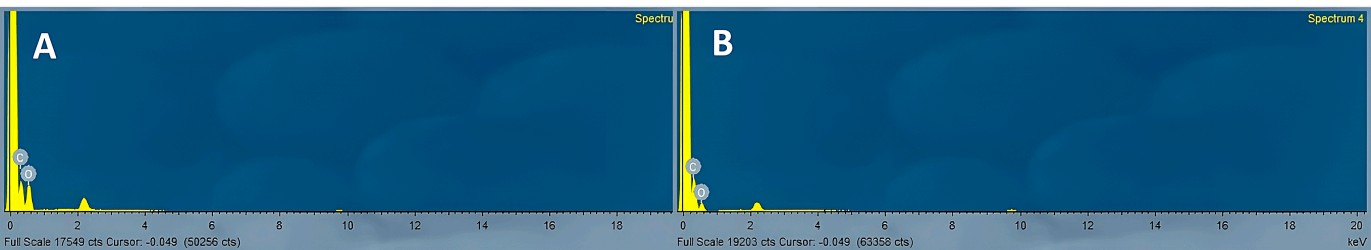

**Figure 5.** EDX analysis of untreated cotton (**A**) and graphite-based cotton (**B**).

### 3.3.2. XPS Analysis

XPS analyses are performed to determine the chemical composition of untreated and graphite-based cotton fabrics. The XPS spectra of these samples are shown in Figure 6. The main elements identified in cellulose fibers are carbon and oxygen (see Figure 6A and Table 2). However, cellulose also contains hydrogen atoms, but these cannot be detected by XPS because they do not have core electrons. The oxygen–carbon atomic ratio was calculated using area sensitivity factors and was 0.52, which is lower than the theoretical value of cellulose. This could be due to the fact that cotton fibers consist of $\alpha$-cellulose (88.0–96.5%) in addition to noncellulosic components. When comparing our results with those of older studies, it should be noted that our O/C ratio for untreated cotton is higher than the O/C ratio reported by Mihailovic et al. [50] (O/C = 0.45) and Inbakumar et al. [51] (O/C = 0.30) for cotton fibers. The results of the XPS analysis of the graphite-treated cotton fabrics (see Figure 6B and Table 3) showed a similar spectral pattern, but the C1s signal was significantly higher than the C1s signal of the untreated cotton due to the amount of graphite in the sample, which also leads to a decrease in the O/C ratio to 50.26%.

### 3.4. XRD Analysis

The XRD patterns of untreated and graphite-based cotton fabrics are shown in Figure 7. The characteristic peak of the untreated cotton (Figure 7A) was at 14.24°, 16.49°, 22.58°, and 34° and corresponds to the lattice spacings of 6.22 Å, 5.29Å, 3.98 Å, and 2.64 Å, respectively. The result of this analysis is then compared with the XRD of the graphite-based cotton fabrics (Figure 7B), which shows two things. First, a sharp peak at 26.45° and a peak at 54.49° were observed, which are assigned to graphite and correspond to lattice spacings of 3.369 Å and 1.68 Å, respectively. Second, the peaks of the untreated cotton fabrics were also observed in this pattern. The XRD results shown in our study are consistent with the results of other studies published in the literature for cotton and graphite [43].

**Table 2.** Peak table of untreated cotton fabrics.

| Name | Start BE | Peak BE | End BE | Height CPS | FWHM (eV) | Area (P) CPS. ev | Area (N) KE^0.6 | Atomic % |
|---|---|---|---|---|---|---|---|---|
| C1s | 292.58 | 286.19 | 277.08 | 2104.88 | 4.83 | 11,097.32 | 157.62 | 65.66 |
| O1s | 541.08 | 533.09 | 523.08 | 3366.3 | 3.5 | 14,808.09 | 82.42 | 34.34 |

**Table 3.** Peak table of graphite-treated cotton fabrics.

| Name | Start BE | Peak BE | End BE | Height CPS | FWHM (eV) | Area (P) CPS. ev | Area (N) KE^0.6 | Atomic % |
|------|----------|---------|--------|-----------|-----------|-----------------|----------------|----------|
| C1s | 289.58 | 285.17 | 277.08 | 18,277.47 | 3.04 | 61,537.43 | 873.58 | 66.55 |
| O1s | 552.52 | 543.63 | 527.58 | 5850.34 | 13.21 | 78,364.25 | 439.07 | 33.45 |

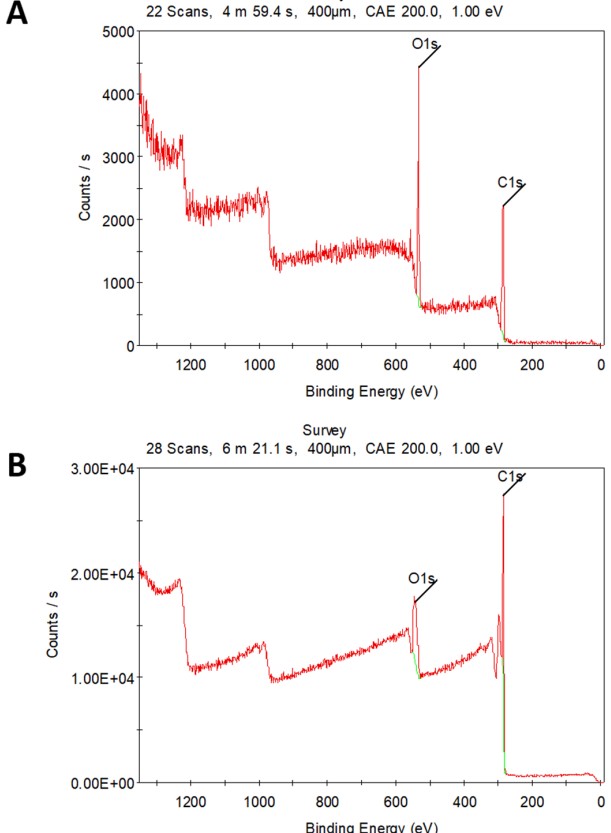

**Figure 6.** XPS analysis of untreated cotton (**A**) and graphite-based cotton (**B**).

*3.5. Graphite-Based Cotton Fabrics for Capacitive Touchscreen Devices*

Capacitive touchscreens are commonplace for most people and can be found in ATMs, cell phones, and laptops. In addition, most restaurants and markets have self-service systems that allow customers to order and pay by touchscreen. These screens consist of multiple layers of glass and plastic coated with a conductive material that responds when the screen is touched with another electrical conductor, such as a bare finger. One of the problems with capacitive touchscreens is that they are not compatible with normal gloves because gloves have an insulating effect. To solve this problem, gloves could be equipped with conductive fingertips to make good contact with the touchscreen, but this would increase manufacturing costs. Here we fabricate a smart, durable glove for touchscreen devices that incorporates a small amount of conductive material in one of the fingertips. A simple experimental procedure was used as follows: First, we prepared graphite dispersion in ethanol; then we placed a drop of the solution onto one of the fingertips and left it for 15 min. We then dried the glove at 100 °C for 30 min. As shown in Figure 8, our smart glove was successfully used on a smartphone touchscreen and also on self-service devices. This means that good electrical contact was established between the fingertip of the glove and the touchscreen. In addition, the durability of our smart glove was tested for up to 10 wash cycles. The result showed that our smart glove remained conductive up to 10 washing cycles and can be used in touchscreens without any problems. Kim

et al. fabricated a conductive para-aramid knit using a hybrid of graphene and waterborne polyurethane [52]. The dip-coating technique was used to fabricate the conductive layer, which can achieve up to five coating cycles. The sample with five coatings worked only on a smartphone touchscreen, while other samples with one to four coatings did not work. When comparing our results with those of older studies, it should be emphasized that we have achieved good results with a simple and durable method using a smaller amount of graphite. Our approach is simple, environmentally friendly, and cost-effective compared with other methods. One of the key benefits is that these smart gloves can be used as personal protective equipment to avoid touching self-service devices when leaving the house during the COVID-19 pandemic. In addition, these smart gloves can be useful for people in harsh environments to operate touchscreen devices. Of course, among the limitations of the present studies is that a small amount of graphite is lost when touching the screen, which contaminates the screen and requires cleaning.

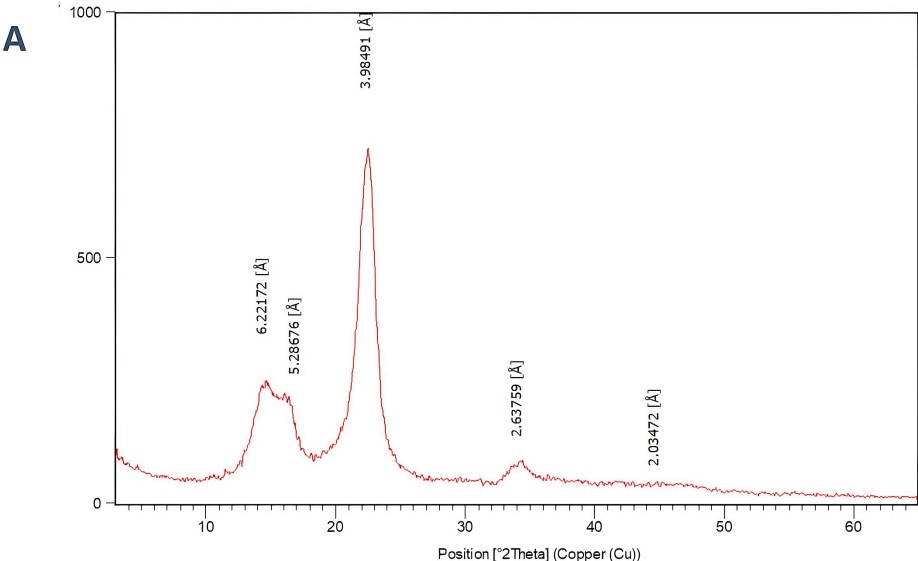

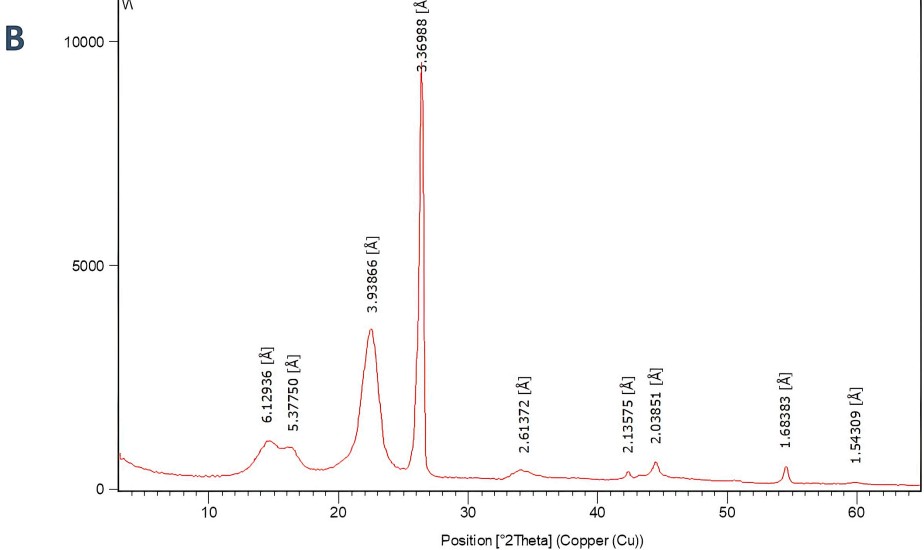

**Figure 7.** XRD patterns of untreated cotton (**A**) and graphite-based cotton (**B**).

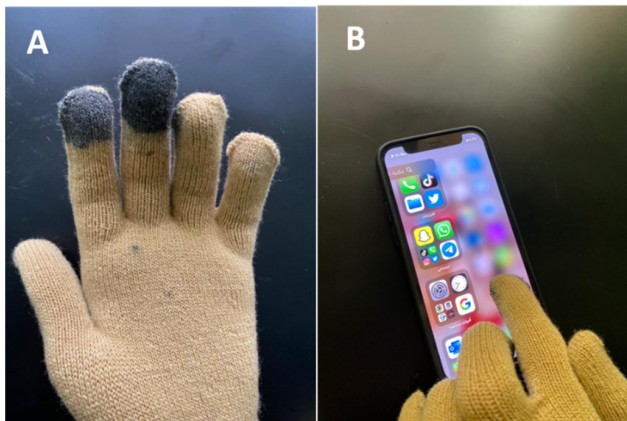

**Figure 8.** (**A**) Digital photograph of a graphite drop-casted glove. (**B**) Coating on the fingertip of the glove can be detected well by the touchscreen of the cell phone.

## 4. Conclusions

In this study, we prepared conductive cotton fabrics with graphite dispersed in ethanol and incorporated the graphite into the cotton by the drop-casting process. The amount of graphite particles was increased by repeating the drop-casting and drying until a saturation concentration of 74.48 wt% was reached, which gave good conductivity and a low sheet resistance of 7.97 $\Omega/\square$. To our knowledge, this is the lowest sheet resistance reported in the literature for graphite-based cotton fabrics. Overall, our results show a strong effect of temperature on the resistivity of graphite-based cotton fabrics, with a transition from semiconductor to metal observed at a transition temperature of 90 °C. These results offer a potential approach for making smart and durable gloves by incorporating the dispersed graphite into one of the fingertips used for touchscreen devices to protect people during the COVID-19 pandemic and also to help people in harsh environments. Moreover, compared with other techniques, this manufacturing process is a simple, eco-friendly, safe, and cost-effective method of producing these smart fabrics. Further research is needed to improve the durability of graphite on cotton fabrics.

**Author Contributions:** Conceptualization, F.A.A.; Resources, W.A.; Data curation, W.A.; Writing—original draft, F.A.A.; Writing—review& editing, F.A.A.; Supervision, F.A.A. All authors have read and agreed to the published version of the manuscript.

**Funding:** This research received no external funding.

**Conflicts of Interest:** The authors have no conflict of interest to disclose.

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
