# Peer review of "Eco-Friendly, Low-Cost, and Flexible Cotton Fabric for Capacitive Touchscreen Devices Based on Graphite"

_crystals, doi:10.3390/cryst13030403_

Round 1

Reviewer 1 Report

The work is interesting, but the following should be corrected and added:
1. The "minimum value of 7.97 Ω/□" in lines 59, 77, 124, 126, 127, 129, fig. 2, 246 should be corrected.
2. Check the author's last name Scha * et al, line 66
3. It says: in line 91 "and then stirred ultrasonically" and in Fig.1 "Magnetic stirrer",
4. The electrical resistance of the samples should be better explained or iliterature references should be provided, line 115-118
5. Measurements on textiles are made under standard conditions of temperature and relative humidity, this is not stated in the paper and under what conditions this was measured ... Pamuk is very hygroscopic and this data can influence the results.
6. In lines 80-83 it states, "Based on the distinct Electrical properties, we produce smart Gloves for touchscreen devices, in which a small amount of graphite dispersion has been used as a conductive material in the fingertips of the Gloves to establish Electrical Contact between our smart Gloves and touchscreen devices."
It should be explained why samples were tested at temperatures up to 130 °C, even though the temperature at the top of the glove did not exceed 40 °C.
7. The exact conditions for 10 wash cycles should also be stated, as they are highly dependent on the durability of the cotton deposits.

Author Response

Response to reviewer #2

Thank you for your comments and your time to read our manuscript. We know that you have spent a great deal of time reviewing this article, and we thank you for your time. We have responded to the comments point by point.

Comment 1: "Eco-friendly" - no eco-data was given and discussed in the paper

Response to comment 1: Thanks for your comment. We mean by the eco- friendly, according to the Cambridge dictionary, we mean "designed to have little or no damaging effect on the environment. Basically, it's all about doing no harm. Products, events and services that are eco- friendly don't cost the earth," and in our manuscript, we make environmentally friendly smart gloves for touchscreen devices. We added the following statement in the conclusion

Moreover, compared to other techniques, this manufacturing process is a simple, eco-friendly, safe, and cost-effective method of producing these smart fabrics.

Comment 2: no cost analysis data was given

Response to comment 2: Thank you for your comment. We added the following statement in the fabrication section:

The manufacturing process was inexpensive due to the cheapness of our raw materials, namely graphite, cotton and ethanol. To calculate the price of our product for a conductive cotton fabric, we need to add the cost of graphite, cotton fabric and ethanol. According to Sigma Aldrich, the price of 1 kg of graphite is $77, and this amount is enough to make a large number of samples because the amount of graphite in each sample does not exceed 100 mg (this is for samples with high concentration). Moreover, to make the smart gloves, we add only a small amount of graphite to one of the fingertips of the glove, which means that this process will be inexpensive.

Comment 3: we know the cotton fabric is flexible.  However, after the graphite treatment, no "flexible" data was given.

Response to comment 3: I thank you for your comment. Regarding the smart gloves, it is obvious that the material is still flexible. However, we are currently working on a new project and will compare the fabrication of conductive cotton fabrics with graphene, its derivative and graphite using different organic solvents and also investigate the mechanical properties.

Reviewer 2 Report

This paper mentioned the topic of "Eco-Friendly, Low Cost and Flexible Cotton Fabric for Capacitive Touch Screening Devices Based on Graphite".  However, the following issues were not fully discussed?

"Eco-friendly" - no eco-data was given and discussed in the paper.

"Low cost" - no cost analysis data was given.

"Flexible" - we know the cotton fabric is flexible.  However, after the graphite treatment, no "flexible" data was given.

Author Response

(The authors gave the same response as above.)

Reviewer 3 Report

First of all the "crystal" aspect of the paper is very low. The authors use graphite powder which is dispersed in ethanol and dipped on a fabric. But has this anything to do with the crystal structure of the material? The authors mention "graphite nanopowder" far from being a crystal. 

The authors use the four line probe technique for electric measurements. The correct nomenclature is the four contact method.

Line 87: a pure cotton fabric.... was this a woven structure? Nitted? How does it look like?

line 127-128: the distance between the voltage probes is only 0.35 cm. What is the distance between two neighbouring yarns of the fabric? It possible that you measure a voltage across a distance which is only 2-3 yarns.

The results shown in fig.3 is a well known phenomenon. It has been extensively discussed in the literature especially in the field of thick film microelectronics. Resistors were made with a carbon+filler paste on a ceramic substrate. It was found that the resistivity decreases toward a certain temperature and then increased again. Physical explanation were also given. 

In line 125 I read that for a concentration of 18% the square resistance is 35 kOhm = 3.5 104 Ohm. But fig.2 tells me 4 103 almost 10 times less. I found several contradictions between the text and the figure. This is totally unacceptable for a scientific paper.

The graph in fig.2 looks like a hyperbola. It would be better to plot it with double logarithmic scale. Then it is obvious that the reverse 1/Rs is just proportional to C which is quite obvious.

figure 7: digital photograph Does it matter what kind of photograph this is? Fig7B shows some part which is unclear without any reason.

several references do not provide the volume number, page number,... reference 14: the journal "A. Physical" does not even exist!

 there are too many shortcomings to make this paper acceptable for publication in a scientific journal.

Line 69: what is the doctors knife method?

Author Response

Response to reviewer #3

Thanks for your comments and your time to read our manuscript. We have responded to the comments point by point.

Comment 1: First of all the "crystal" aspect of the paper is very low. The authors use graphite powder which is dispersed in ethanol and dipped on a fabric. But has this anything to do with the crystal structure of the material? The authors mention "graphite nanopowder" far from being a crystal. 

Response to comment 1: Thank you very much for your comment. We are currently working on a new project and will compare the preparation of conductive cotton fabrics using graphene, its derivative and graphite using different organic solvents and calculate the crystallinity before and after treatment.

Comment 2: The authors use the four line probe technique for electric measurements. The correct nomenclature is the four contact method.

Response to comment 2: Thank you very much for your comment. We changed to four probe method.

Comment 3: Line 87: a pure cotton fabric.... was this a woven structure? Knitted? How does it look like?

Response to comment 3: Thank you for your comment. It’s a pure woven cotton and we fixed the statement as follows: A pure woven cotton fabric was purchased from a local store.

Comment 4: line 127-128: the distance between the voltage probes is only 0.35 cm. What is the distance between two neighbouring yarns of the fabric? It possible that you measure a voltage across a distance which is only 2-3 yarns.

Response to comment 4: I thank you for your comment. This is a good point for research to consider in the future and could be useful for other applications and making conductive yarn using graphite.

Comment 5: The results shown in fig.3 is a well known phenomenon. It has been extensively discussed in the literature especially in the field of thick film microelectronics. Resistors were made with a carbon+filler paste on a ceramic substrate. It was found that the resistivity decreases toward a certain temperature and then increased again. Physical explanation were also given.

Response to comment 5: Thank you for your excellent comment and explanation. We have added the following explanation at your suggestion:

This semiconductor-metal behavior is a well-known phenomenon that has been observed in microelectronics, particularly in the fabrication of resistors with carbon-filler paste on a ceramic substrate. In the study presented by Zha et al.52 on the dependence of resistance on temperature, the semiconductor-metal behavior was observed for the composite with hybrid filler carbon nanotubes/carbon black and polyethylene/polyvinylidene fluoride. They found that the temperature transition depends on the concentration of the filler and the melting point of the polymer. In addition, this behavior is also observed in the manufacture of conductive fabric using conductive polymers, graphene and carbon nanotubes.  Alamer et al.53, 54 demonstrated that the conductive cotton fabrics coated with PEDOT:PSS exhibited semiconductor-metal behavior and the transition temperature depended on the PEDOT:PSS concentration. In another study27, the thermal investigation of the graphene/graphite infused PET fabrics showed that the conductive fabrics exhibited semiconductor metal behavior at 350 K in which the study was conducted in a temperature range from 10 K to 400 K.

Comment 6: In line 125 I read that for a concentration of 18% the square resistance is 35 kOhm = 3.5 104 Ohm. But fig.2 tells me 4 103 almost 10 times less. I found several contradictions between the text and the figure. This is totally unacceptable for a scientific paper.

Response to comment 6: Thanks for your good note. We apologize for this error. We have drawn Figure 2 and added Table 1 showing the raw data of Rs at various graphite concentrations. The correct value is 35. 76 k Ω/□. We apologize for this as well, and believe me, it's my master's student's mistake.

Comment 7: The graph in fig.2 looks like a hyperbola. It would be better to plot it with double logarithmic scale. Then it is obvious that the reverse 1/Rs is just proportional to C which is quite obvious.

Response to comment 7: Thank you for your comment. We plotted the relation between the natural logarithm of sheet resistance and the natural logarithm of concentration. We added the following statement   

Figure 2A shows a hyperbolic curve, and at high graphite concentration it is not clear what the value of sheet resistance is. To determine the relationship between sheet resistance and graphite concentration, we plotted the natural logarithm of sheet resistance and the natural logarithm of graphite concentration (see Figure 2B). It is obvious that the sheet resistance is inversely proportional to the graphite concentration

Comment 8: figure 7: digital photograph Does it matter what kind of photograph this is? Fig7B shows some part which is unclear without any reason.

Response to comment 8:  I thank you for your comment. But yes, it is important. It shows the treatment of the fingers with graphite and is the application of this study. In Figure 7B, it is not an unclear part, but we took the picture during the movement of the iPhone screen to prove the touch screening, and we have attached a video.

Comment 9: several references do not provide the volume number, page number,... reference 14: the journal "A. Physical" does not even exist!

Response to comment 9: Thank you for your comment to look also in the references. We fixed it.

Comment 10: Line 69: what is the doctors knife method?

Response to comment 10: Thank you for your comment. The detailed of this in ref. 49

Reviewer 4 Report

Dear Authors,

In order to improve the readability of your paper I have some comments ad suggestions:

1. Fig2 Rs cannot be negative, the y-scale should start in 0.

2. In order to compare Fig2 and Fig3 the same value should be represented. I suggest to plot sheet resistance on Fig.3

3. The authors are using SEM, EDX, XPS, XRD some reader maybe are not familiarized with all theses techniques, a brief explanation of each will improve the readability of the paper.

BR

Author Response

Response to reviewer #4

Thank you for your comments and your time to read our manuscript. We know that you have spent a great deal of time reviewing this article, and we thank you for your time. We have responded to the comments point by point.

Comment 1: " Fig2 Rs cannot be negative, the y-scale should start in 0.

Response to comment 1: Thanks for your comment. We fixed it and plot it again.

Comment 2: In order to compare Fig2 and Fig3 the same value should be represented. I suggest to plot sheet resistance on Fig.3

Response to comment 2: Thank you very much for your comment. There is no comparison between Figure 2 and Figure 3. In Figure 2, we studied the effect of concentration on the sheet resistance of the conductive cotton fabric. In Figure 3, we studied the effect of temperature on the behavior of the materials, i.e., is the behavior of the material metal or semiconductor?

Comment 3: . The authors are using SEM, EDX, XPS, XRD some reader maybe are not familiarized with all theses techniques, a brief explanation of each will improve the readability of the paper.

Response to comment 3: I thank you for your comment. We have a statement of purpose in each technique. However, we are currently working on a new project and will compare the fabrication of conductive cotton fabrics with graphene, its derivatives and graphite using different organic solvents and also provide more information about SEM, EDX, XPS, XRD to help people from the field understand the story.

Thanks for your comments

Round 2

Reviewer 2 Report

No further comment

Reviewer 3 Report

Thank you for this revised version. It looks OK to me.

Reviewer 4 Report

I recommend to publish the paper in the present form